# A Prospective Randomized Controlled Multicenter Clinical Trial Comparing Paste-Type Acellular Dermal Matrix to Standard Care for the Treatment of Chronic Wounds

**DOI:** 10.3390/jcm11082203

**Published:** 2022-04-14

**Authors:** Youn Hwan Kim, Hyung Sup Shim, Jihye Lee, Sang Wha Kim

**Affiliations:** 1Department of Plastic and Reconstructive Surgery, College of Medicine, Hanyang University, Seoul 04763, Korea; younhwank@daum.net; 2Department of Plastic and Reconstructive Surgery, College of Medicine, The Catholic University of Korea, St. Vincent’s Hospital, Seoul 06591, Korea; shrapshim@catholic.ac.kr; 3CG Bio, Seoul 04349, Korea; jhlee2818@cgbio.co.kr; 4Department of Plastic and Reconstructive Surgery, College of Medicine, Seoul National University, Seoul National University Hospital, Seoul 03080, Korea

**Keywords:** acellular dermis, wound healing, ulcer

## Abstract

The treatment of chronic wounds remains challenging. Acellular dermal matrix (ADM) has been shown to be effective for various types of wound healing. This study was designed to compare the wound size reduction rate after 12 weeks between patients receiving paste-type ADM and standard wound care. Patients over 19 years old with chronic wounds, deeper than full-thickness skin defects, more than 4 cm^2^ in size that did not heal over the 3 weeks before the study were included. After a screening period of 7 days, patients were randomized to receive either paste-type ADM or standard wound care. The wound status was evaluated at baseline, 1, 2, 4, 8, and 12 weeks. A total of 86 patients were enrolled in this study. The wounds continuously and constantly reduced in size from week 1, and the reduction rate was significantly greater in the study group from week 2 until the end (week 12). In the study group, wound healing was achieved in 29 of 38 wounds (76.3%). Paste-type ADM might be a useful option for wound healing and can be applied safely and efficiently for advanced wound care.

## 1. Introduction

Wound healing progresses systematically through inflammation, proliferation, and remodeling phases [1,2]. Interference in this well-coordinated process, especially in the inflammatory stage, leads to chronic non-healing wounds [2]. Chronic wounds often occur in patients with comorbidities such as diabetes, vascular problems (including arterial disease and venous ulcers), or chronic inflammation (such as osteomyelitis, autoimmune disease, and radiation ulcers) [3,4,5,6]. It is estimated that 1–2% of the population of developed countries experience chronic wounds [5], which not only affect quality of life but also increase healthcare costs [3,7,8].

The treatment of chronic wounds remains challenging. When these wounds are non-responsive to conventional wound management modalities, advanced wound care materials, including cultured autologous material, allogenic materials, and bioengineered products, are required to accelerate wound healing [3,9,10,11,12].

Acellular dermal matrix (ADM) is a biomaterial derived from autologous and allogenic tissues that undergoes processing to remove cells, while still retaining the bioactive dermal matrix, consisting of collagen, elastin, and fibronectin [2,3,4,7]. ADM has been used widely in various applications as a dermal replacement, including for the head and neck, breast, abdominal wall, and extremity reconstruction, and has also been shown to be effective for tissue regeneration and wound healing [2,3,4,7,12,13,14].

Although ADM is commonly used in sheet form, paste-type ADM manufactured by crushing and micronizing allograft material derived from donated human skin has recently been introduced. This transformation makes it is easy to handle and paste-type ADM can be applied to various types of wounds, including external, ulcerative and irregularly shaped tunneling wounds [7,12,13,14,15]. Previous studies have demonstrated the efficacy of sheet-type ADM as a dermal substitute for various types of wound reconstruction [2,4]. Other studies have reported the outcomes of ADM therapy for chronic wounds, such as diabetic foot ulcers [10,11,12,16,17]. However, few studies have used paste-type ADM, and most were not randomized controlled studies.

The primary objective of this study was to compare the wound size reduction rate after 12 weeks between patients receiving ADM therapy and standard wound care. The secondary objectives were to compare the complete wound healing rate, the epithelization rate, granulation tissue formation, and safety.

## 2. Materials and Methods

This was a 12-week prospective randomized controlled multicenter clinical trial conducted to determine the efficacy and safety of paste-type ADM therapy. This study was approved by the institutional review boards of Seoul National University (1704-063-845), Hanyang University (2017-01-061), and the Catholic University of Korea (VC17DNSI0079). All participants provided written informed consent for the publication of the case details, including images. The study was registered at ClinicalTrials.gov (NCT04019639). All of the data were analyzed anonymously and in accordance with the principles of the 1975 Declaration of Helsinki (revised in 2008).

Two products were used in this study: CG-PASTE (CG Bio Co., Ltd., Seoul, Korea) and Easyfoam (CG Bio Co., Ltd., Seoul, Korea). CG-PASTE is a paste-type micronized acellular dermal matrix that is currently used safely in clinical practice; it has been approved as a medical device applicable to open wounds except for third-degree burns. Easyfoam is a wound dressing applied to wounds with exudates and protects wounds by maintaining a moist environment on the wound.

Each patient was screened for eligibility, including a complete medical history, physical examination, and full assessment of the wound, based on the inclusion and exclusion criteria shown in Table 1.

Patients > 19 years old, with chronic wounds with the wound depth ranging from full-thickness skin to bone exposure measuring more than 4 cm^2^ and failing to heal during a minimum of 3 weeks before the study [9,10,15], were eligible for inclusion.

The exclusion criteria were superficial or partial-thickness skin defects, undermining or tunneling wounds for which it was difficult to measure the wound depth, and uncontrolled infection, including osteomyelitis. Patients with poor metabolic control (HgA1c > 12% within the previous 3 months), a serum creatinine level > 3.0 mg/dL, treatment with other medical devices or topical growth factors that can influence wound healing within the previous 30 days were excluded.

Patients were evaluated during a screening period up to 7 days before baseline surgical debridement and treatment. All patients underwent debridement until healthy, viable tissue was visible in the wounds. After the surgical preparation of the wound site, patients were randomized into either the experimental group (paste-type ADM and conventional dressing) or control group (conventional dressing) using sequentially numbered, opaque, sealed envelopes to avoid selection bias.

Wounds were assessed at 0 (baseline, after initial surgical preparation of wound site), 1, 2, 4, 8, and 12 weeks after randomization, and upon study exit or withdrawal. Photographs of the wounds were taken at a distance of 30 cm. A centimeter scale was placed adjacent to the wound. Wound size, granulation tissue formation, epithelization, complete healing status, and adverse events were recorded at each follow-up visit.

For patients assigned to the experimental group, paste-type ADM (CGPaste) was placed on the wound bed to cover the entire wound surface, and then covered with polyurethane foam (EasyFoam). It was applied at 0, 1, 2, and 4 weeks after the initial surgical preparation of the wound. For patients in the control group, wounds were covered with conventional dressing using polyurethane foam only.

The main outcome was wound size reduction over the 12-week follow-up period. Secondary outcomes were the achievement of complete healing (defined as an epithelized wound with no raw surface and no requirement for additional wound management), time to complete healing, and granulation tissue formation during the follow-up period. Wound granulation and epithelialization were evaluated by using photographs of the wounds taken under similar conditions (distance, brightness, etc.). The evaluation of photographs was performed by two independent evaluators who are experts in the plastic surgery department. Granulation tissue formation was evaluated as the percentage of the wound surface that was covered with bright-red healthy granulation tissue. The granulation rate was defined as the percentage of patients who achieved over 75% granulation. Epithelialization is defined as the wound covered with an epithelial surface. The epithelialization rate was calculated as the percentage of patients who achieved epithelization. Adverse events, including wound infection or any complications, were also evaluated.

### Statistical Analysis

Categorical variables were analyzed using contingency tables (Chi-square) and continuous variables were analyzed using either the independent *t*-test or Mann–Whitney test, depending on whether the data met the criteria for parametric analysis. The times to healing and granulation were compared between the two treatment arms using the log-rank test. A statistical analysis was performed using GraphPad Prism for Windows (version 5; GraphPad Software Inc., San Diego, CA, USA), and a two-sided *p* < 0.05 was taken to indicate statistical significance.

## 3. Results

A flow chart of study enrollment and participation through the clinical trial is shown in Figure 1.

According to the literature, the chronic wound area of the control group after 12 weeks was expected to decrease by about 92.3% compared to the baseline [18], and that of the test group’ was expected to decrease by about 98% [19]. Therefore, in this study, it was assumed that reasonable clinical improvement was reached when the difference between the groups in wound area reduction rate was 6% or more (80% power and 5% significance level). The sample size for this clinical trial was calculated as 84 patients (42 patients per group), taking into consideration a 15% drop-out rate.

Of the 86 patients enrolled in the study, five were considered screening failures. The remaining 81 patients were randomized into two treatment groups, with 41 patients (42 wounds) receiving paste-type ADM (study group) and 40 (40 wounds) receiving standard care (control group). Eight patients (four in the study group and four in the control group) did not complete the clinical trial.

Table 2 shows a comparison of demographic characteristics, including the initial wound size between the two groups upon enrollment. The two groups were comparable in terms of age, sex, and comorbidities (including diabetes, hypertension, autoimmune disease, and vascular disease). The baseline wound size was also not significantly different between the groups.

The wound size reductions at the evaluation points in both treatment groups are presented in Table 3. The wounds showed a continuous and relatively constant reduction from week 1 in the study group and week 4 in the control group (Figure 2). There was a significant difference in the wound area reduction rate between the groups from week 2 to the study endpoint (week 12).

The percentage of granulation tissue in the wound (Figure 3), as well as the wound epithelization rate (Figure 4), showed substantial increases over time. The differences between the groups were clinically significant for both parameters (*p* = 0.0006 and *p* = 0.0016, respectively).

Wound healing was defined as complete epithelization without a raw surface. Figure 5 shows the percentage of wounds that healed completely over the course of treatment. In the study group, 29 of 38 wounds (76.32%) were healed by 12 weeks, compared to only 11 of 36 (30.56%) in the control group (*p* = 0.001).

No adverse events were noted during treatment.

### 3.1. Case 1

A 27-year-old female patient presented with a third-degree contact burn on her lower left leg. After debridement, the wound measured 6.0 × 4 cm (Figure 6). We applied 2 cc of paste-type ADM and covered the wound with polyurethane foam dressing. After 4 weeks, the wound size had reduced by approximately 50%, and it had healed almost completely after week 8, but the epithelialized wound appeared “scratched” due to trauma. Complete healing was observed by week 12.

### 3.2. Case 2

An 81-year-old male patient presented with an open wound on his lower left leg due to trauma. The patient was treated at a local clinic for over 1 month, but the wound did not heal. A 5 × 4 cm skin defect was observed after debridement (Figure 7). We applied 2 cc of paste-type ADM with a polyurethane foam dressing. Paste-type ADM was reapplied at 1, 2, and 4 weeks. At week 8 after initial treatment, the wound size had reduced to 1.5 × 2 cm. After 12 weeks, the wound had healed completely. No contracture deformity was observed and a good esthetic outcome was achieved.

### 3.3. Case 3

A 77-year-old male patient presented with a diabetic foot ulcer in the right lateral malleolus region. A rotation flap was applied from the foot dorsum and skin grafting was performed at the donor site. However, the flap was necrotized and the skin graft only partially took. After debridement, an open wound measuring about 5 × 3 cm was observed in the lateral malleolus region and on the foot (Figure 8). After the application of paste-type ADM at weeks 0, 1, and 2, the wound on the foot healed almost completely and the wound on the lateral malleolus was covered with healthy granulation tissue. The wound gradually reduced in size, and showed complete healing by week 12.

The patient was satisfied that the wound had healed completely without surgery, with no discomfort in the foot or ankle and normal function.

## 4. Discussion

Wound healing is a well-coordinated process involving interactions of cells with the microenvironment. The extracellular matrix (ECM) is one of the key elements in wound healing, providing structural support as the largest component of the dermal layer [2,15] and also promoting effective wound healing by providing signaling proteins for cell adhesion and signaling [10,20,21,22]. As the ECM is often dysfunctional or insufficient in chronic wounds, it is challenging to promote wound healing. Efforts have been made to replace the damaged ECM or restore its function to stimulate wound healing [2,3]. The application of ADM has been applied as an alternative for the ECM in chronic wounds [15,23,24].

The use of ADM provides several advantages. First, the ADM undergone processing to remove cellular components, which makes it immunologically inert [3,4,7,12]. Second, the ADM scaffold comprised of collagen, elastin, and fibronectin provides a favorable microenvironment for cellular proliferation and vascularization [7,12,25]. Third, by retaining the function of the ECM in cell adhesion and cell signaling, ADM promotes fibroblast attachment, attracts vascular endothelial cells, and helps growth factors [15,26,27,28]. These properties allow the initiation of self-regeneration processes of wound healing in chronic non-healing wounds [14,29]. Interactions between the surrounding tissue and ADM could result in wound healing by re-epithelialization or granulation tissue formation [14].

This study used paste-type ADM, which is crushed, micronized, and packed in a syringe, and can therefore be applied more easily than sheet-type ADM. Most of the participants in this study were enrolled on an outpatient basis. In addition, it can be applied to complicated wounds, including deep wounds, cavitary wounds, undermining and tunneling wounds, wounds with dead space, etc. [7,30,31]. Therefore, paste-type ADM is particularly useful for sores and diabetic ulcers, where it can be adjusted to maximize contact with irregular surfaces.

The clinical efficacy of a paste-type ADM for various wounds has recently been studied in animal experiments and clinical trials. However, there has been little progress in clinical studies and most such reports to date have been case studies. Paste-type ADM has shown comparable effectiveness for wound healing to subcutaneous injection in rat models [14]. Several authors have reported the clinical efficacy of the ADM application for wound coverage. Early retrospective case studies suggested that ADM may promote wound healing without surgery [7,32]. Jeon and Kim performed a retrospective study with application of ADM to chronic wounds, and wound healing was achieved in 2.4 weeks on average in five out of seven cases [30]. Ahn et al. performed a prospective clinical trial of 20 patients using paste-type ADM in conjunction with negative pressure wound therapy (NPWT), achieving a wound area reduction rate of 59.1% after 4 weeks [15]. In a single-center randomized controlled trial, Brigido compared the application of human ADM to conventional methods using gauze dressing in uninfected wounds of the lower extremities and showed that 12 of 14 patients (85.7%) in the ADM group achieved complete healing after the 16 weeks of treatment, in comparison to only 4 of 14 patients (28.7%) in the control group [7]. Hahn et al. performed a prospective randomized pilot study of micronized human ADM in 30 patients with diabetic foot ulcers, comparing the application of ADM to conventional NPWT, and reported that 93.3% of patients in the experimental group and 85.7% in the conventional therapy group achieved complete wound healing during the 6-month follow-up [31].

The results of this prospective randomized multicenter study indicated that chronic wounds treated with ADM have a higher probability of healing compared to those treated with conventional management. This study had two main goals. The first was to compare the wound size reduction rate after 12 weeks between patients receiving paste-type ADM and standard wound care, and the second was to compare the epithelization rate, growth of granulation tissue, complete wound healing rate, and safety between treatment groups. Immediately after surgical preparation and the application of ADM, the wound size continuously and constantly reduced from week 1, and from week 2, the wound size reduction was significantly greater in the study group until the end (week 12). Furthermore, granulation tissue formation was observed in 36 of 38 wounds (94.7%) in the study group compared to 26 of 36 wounds (72.2%) in the control group, while full epithelization was observed in 34 of 38 wounds (89.5%) in the study group and 18 of 36 wounds (50%) in the control group. Consequently, wound healing was achieved in 29 of 38 wounds (76.3%) in the study group compared to 11 of 36 wounds (30.6%) in the control group. In addition, no adverse events were noted during treatment.

Although debridement is not the standard of care for wound management, it was performed in this study to ensure that the two treatment groups were treated equally, and to promote wound healing while minimizing differences by wound type. The failure of wound healing is often caused by a prolonged inflammatory phase or poor vascularization. Debridement reduces the bacterial burden, removes biofilms and necrotic tissue, ensures viable cells at the wound edges, and helps prepare the wound bed before wound management. We assumed that the appropriate preparation of the wound bed improved the wound healing process in both treatment groups, resulting in complete wound healing in 40 of 74 wounds. There were no adverse events during treatment, including infection, indicating that the process was performed carefully in a sterile environment, and that the paste-type ADM infection rate is low such that it can be used safely.

The strengths of this study include the prospective randomized controlled design and involvement of multiple centers. Although 12 weeks is a common endpoint in wound studies, a longer follow-up would have been beneficial. In addition, there were no significant differences in demographic characteristics, including initial wound size, between the treatment groups, such that the outcomes reflected only the effects of the different treatments. The inclusion criterion was chronic non-healing wounds, with a depth from full-thickness skin defects to bone exposure, and a size measuring more than 4 cm^2^ that failed to heal during a minimum of 3 weeks of conservative care before the study. As various factors can lead to chronic wounds, it is difficult to characterize a wound only according to location or type. Therefore, wound depth, size, and duration were considered in this study.

Regarding study limitations, paste-type ADM application was applied with polyurethane foam dressing in all cases. There may be differences in the effects of ADM according to the dressing material used at the same time. In addition, paste-type ADM was not compared to other types of conventional dressing, such as gauze or NPWT, or to other types of ADM.

This prospective randomized multicenter study showed that treatment of chronic wounds with ADM reduced wound size, increased the epithelization rate and granulation tissue formation, and achieved a higher rate of complete wound healing compared to conventional management. Therefore, paste-type ADM might be a useful option for wound healing and can be used safely and efficiently for advanced wound care.

## Figures and Tables

**Figure 1 jcm-11-02203-f001:**
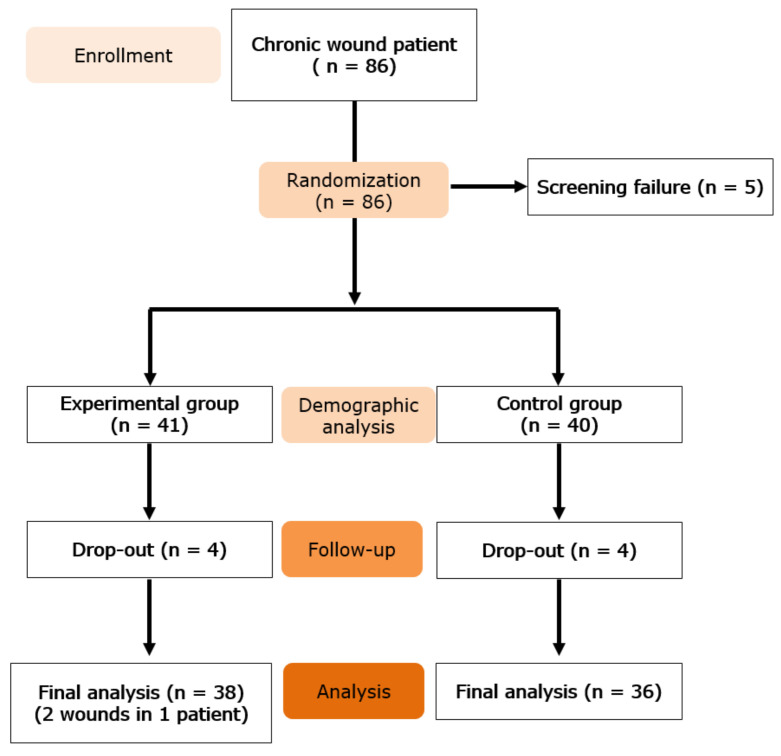
Flow of study enrollment and participation.

**Figure 2 jcm-11-02203-f002:**
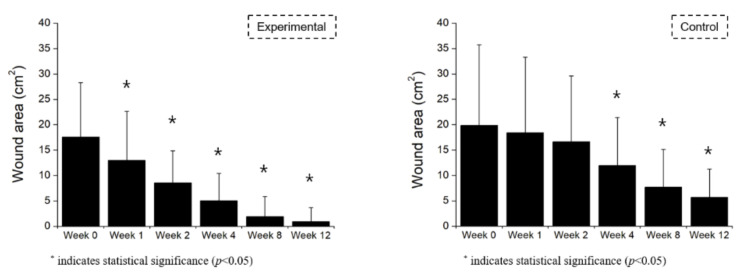
Wound area reduction by week.

**Figure 3 jcm-11-02203-f003:**
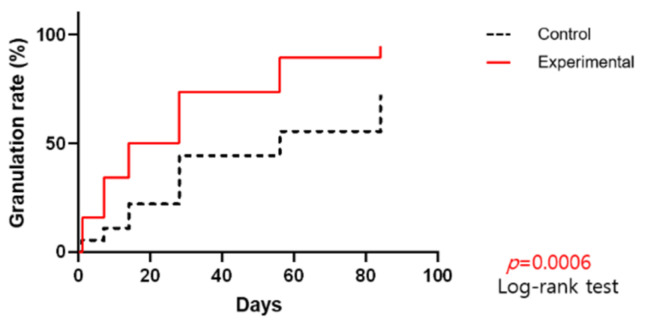
Percentage of granulation tissue in the wound by day.

**Figure 4 jcm-11-02203-f004:**
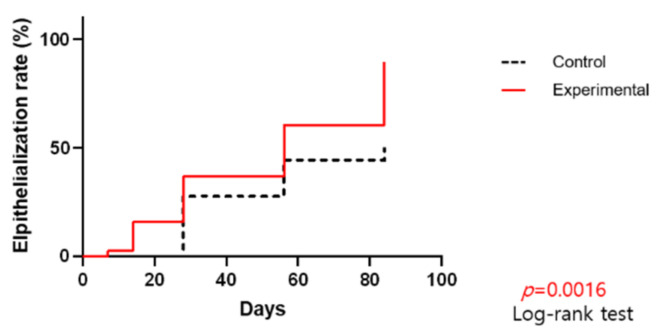
Wound epithelization rate by day.

**Figure 5 jcm-11-02203-f005:**
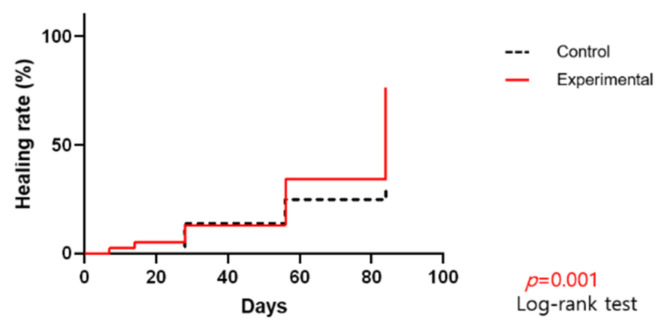
Percentage of wounds completely healed throughout the course of treatment.

**Figure 6 jcm-11-02203-f006:**
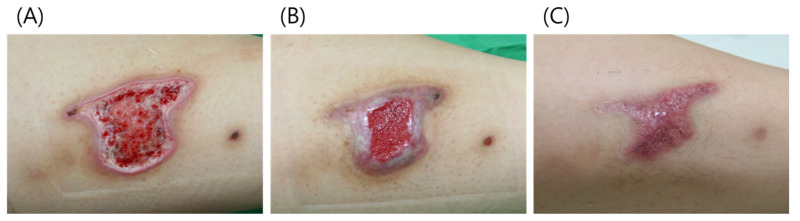
(**A**) A 27-year-old female patient presented with a third-degree contact burn on her lower left leg. (**B**) After 4 weeks, the wound size had reduced by approximately 50%. (**C**) At week 12, the wound had healed completely.

**Figure 7 jcm-11-02203-f007:**
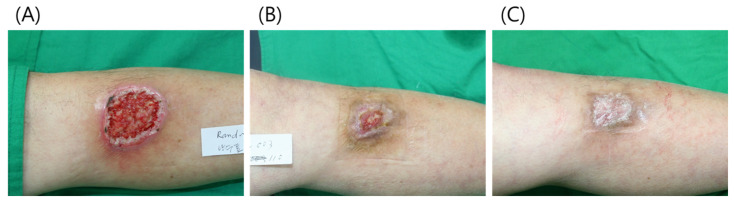
(**A**) A 52-year-old male patient presented with a 5 × 4 cm skin defect on his lower left leg. (**B**) At 8 weeks after initial treatment, the wound size had reduced to 1.5 × 2 cm. (**C**) At 12 weeks, the wound was completely healed without contracture deformity.

**Figure 8 jcm-11-02203-f008:**
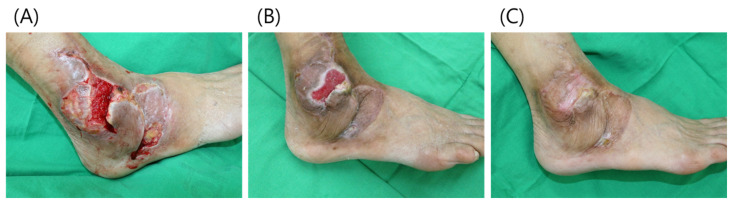
(**A**) A 65-year-old male patient presented with a diabetic foot ulcer. (**B**) After 4 weeks, the wound on the foot had healed almost completely and the wound at the lateral malleolus was covered with healthy granulation tissue. (**C**) At week 12, the wound had healed completely without discomfort in the foot or ankle; normal function also returned.

**Table 1 jcm-11-02203-t001:** Inclusion and exclusion criteria.

Inclusion Criteria	Exclusion Criteria
Patients > 19 years old Full-thickness skin defects to bone exposure wounds Wounds measuring > 4 cm2 Wounds failing to heal following a minimum of 3 weeks of conservative care prior to the study Wounds without uncontrolled infection HbA1c ≤ 12% within the previous 3 months Serum creatinine ≤ 3.0 mg/dL	Superficial or partial thickness skin defects Undermining or tunneling wounds Wounds with uncontrolled infection HbA1c > 12% within the previous 3 months Serum creatinine > 3.0 mg/dL Treatment with other medical devices or topical growth factors within the previous 30 days

**Table 2 jcm-11-02203-t002:** Demographic characteristics.

		Experimental Group (N = 41)	Control Group (N = 40)	*p*-Value
**Age (Year)**	**Mean ± SD**	58.71 ± 16.33	63.63 ± 13.47	0.1436
**Sex**	**Male, N(%)**	23 (56.1)	19 (47.5)	0.4415
	**Female, N(%)**	18 (43.9)	21 (52.5)	
**Smoking**	**No, N (%)**	34 (82.9)	28 (70.0)	0.1735
	**Yes, N (%)**	7 (17.1)	12 (30.0)	
**Alcohol consumption**	**No, N (%)**	33 (80.5)	30 (75.0)	0.5541
	**Yes, N (%)**	8 (19.5)	10 (25.0)	
**Diabetes**	**No, N (%)**	28 (68.3)	24 (60.0)	0.4388
	**Yes, N (%)**	13 (31.7)	16 (40.0)	
**Hypertension**	**No, N (%)**	31 (75.6)	24 (60.0)	0.1351
	**Yes, N (%)**	10 (24.4)	16 (40.0)	
**Hemodialysis**	**No, N (%)**	37 (90.2)	38 (95.0)	0.4131
	**Yes, N (%)**	4 (9.8)	2 (5.0)	
**Vascular disorder**	**No, N (%)**	36 (87.8)	36 (90.0)	0.7543
	**Yes, N (%)**	5 (12.2)	4 (10.0)	
**Wound area (cm^2^)**	**Mean ± SD**	18.01 ± 10.62	19.10 ± 15.37	0.7108

**Table 3 jcm-11-02203-t003:** Wound area reduction.

	Experimental Group	*p*-Value (vs. Baseline)		Control Group	*p*-Value (vs. Baseline)	*p*-Value (Ctrl vs. Exp.)
Baseline (*n* = 41)	18.01 ± 10.62	-	Baseline (*n* = 40)	19.10 ± 15.37	-	0.7108
Week 0 (*n* = 38)	17.57 ± 10.73	0.8544	Week 0 (*n* = 36)	19.86 ± 15.89	0.8328	0.4677
Week 1 (*n* = 37)	13.00 ± 9.63	* 0.0319	Week 1 (*n* = 36)	18.40 ± 14.89	0.8411	0.0692
Week 2 (*n* = 38)	8.56 ± 6.31	* <0.0001	Week 2 (*n* = 36)	16.65 ± 12.93	0.4572	* 0.0009
Week 4 (*n* = 38)	5.05 ± 5.38	* <0.0001	Week 4 (*n* = 36)	11.96 ± 9.46	* 0.01	* 0.0002
Week 8 (*n* = 38)	1.92 ± 3.96	* <0.0001	Week 8 (*n* = 36)	7.69 ± 7.43	* 0.0001	* 0.0001
Week 12 (*n* = 38)	0.90 ± 2.77	* <0.0001	Week 12 (*n* = 36)	5.69 ± 5.58	* <0.0001	* <0.0001

* indicates statistical significance (*p* < 0.05).

## Data Availability

The data presented in this study are available on request from the corresponding author.

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
