# Peer review of "A Prospective Randomized Controlled Multicenter Clinical Trial Comparing Paste-Type Acellular Dermal Matrix to Standard Care for the Treatment of Chronic Wounds"

_jcm, 2022, doi:10.3390/jcm11082203_

Round 1

Reviewer 1 Report

It is a well-written manuscript. But the novelty of the study does not exist. 

There are studies in literature demonstrating that ADM is good for chronic wounds. The authors practically confirm what other studies already described. 

My concern is that in the study are less than 100 patients and the statistic is not completely relevant. 

The study is good but more patients should be included to be relevant for the future literature. 

Author Response

Thank you for the comment.

We evaluated the required number of patients to compare the wound size reduction rate after 12 weeks of treatment. Minimum requirement was 78 patients. Therefore, we enrolled 86 patient, considering screening failure or droup out (10%).

Reviewer 2 Report

The paper entitles “A prospective randomized controlled multicenter clinical trial 2 comparing paste-type acellular dermal matrix to standard care 3 for the treatment of chronic wounds” by Youn Hwan Kim,Hyung Sup Shim ,Jihye Lee,Sang Wha Kim is well written and has some minor corrections for which a few are illustrated in specific comments.

Line 34. Wound healing progresses instead of wound healing progress

Line 40. Experiences and not experiences (population may be taken as plural)

Line 40. Quality-of-life

Line 48. Not really the relevant references.  Are there any references that are directly related to ADM used in the current clinical study.  More specific information would allow to easily follow the manuscript.

Line 56.  Again, the references do not relate to the phrases and it would be good to assure that other studies and preparations are explained so that there is a clear understanding of the formulation used for each study.

Materials and Methods

Begin with a summary of the ADM preparation so that it is clear what the formulation of test is composed of, how it was composed (whether all under GMP or in a transplantation program) and under what regulatory (i.e. medical device etc.). Information on ADM (CGPaste; CGBio, 97 Hwaseong, Korea and its registration with a short summary of formulation would be appropriate to include.

Also, the same for EasyFoam, CGBio- whether this is a registered product and could be labelled as such.

Patient

It could be more clear if this is a continuation of the study published in Archives of Plastic Surgery by Jeon and Kim for a retrospective study using the CGPaste between 2017 and 2018.  When this study was accepted, was it considered as a Phase I safety study for the 7 patients.  If so this study could be better described as a Phase I/II study with more patients assessed by the need of the >80 to achieve statistical difference for diverse chronic wounds. Show the statistical assessment that was done for determining the inclusion number.

Granulation evaluation will need to be described with the full method of evaluation

Epithelization will need to be described with the full method of evaluation

Results

Each Case report could be presented in one figure with the photos shown together over time.

Figures 6, 7 and 8 combined

Figures 9, 10, and 11 combined

Figures 12, 13, 14 combined

Or to put all the patient data in one total figure as this would be easily illustrated this way.

Discussion

Throughout the discussion, the generic use of ADM is not advisable, as each formulation should be described for each study.  Not all formulations are equal and these will need to be described for each study referenced in the discussion.

Author Response

Line 34. Wound healing progresses instead of wound healing progress

-We changed the word

Line 40. Experiences and not experiences (population may be taken as plural)

-We changed the word

Line 40. Quality-of-life

-We changed the word

Line 48. Not really the relevant references.  Are there any references that are directly related to ADM used in the current clinical study.  More specific information would allow to easily follow the manuscript.

-The reference is related to next paragraph, which explains various application of ADM.

Line 56.  Again, the references do not relate to the phrases and it would be good to assure that other studies and preparations are explained so that there is a clear understanding of the formulation used for each study.

-Reference 12-16 are the studies using ADM.

Materials and Methods

Begin with a summary of the ADM preparation so that it is clear what the formulation of test is composed of, how it was composed (whether all under GMP or in a transplantation program) and under what regulatory (i.e. medical device etc.). Information on ADM (CGPaste; CGBio, 97 Hwaseong, Korea and its registration with a short summary of formulation would be appropriate to include.

Also, the same for EasyFoam, CGBio- whether this is a registered product and could be labelled as such.

-Thank you for the comment. We included the information of the products in Materials and Methods section.

- Two products are used in this study; CG-PASTE (CG Bio Co., Ltd., Seoul, Republic of Korea) is paste-type micronized acellular dermal matrix that is currently used safely in clinical practice. This product has been approved as a medical device applicable to open wounds except for 3-degree burns. Easyfoam (CG Bio Co., Ltd., Seoul, Republic of Korea) is a wound dressing applied to wounds with exudates and protects wounds by maintaining a moist environment on the wound.

Patient

It could be more clear if this is a continuation of the study published in Archives of Plastic Surgery by Jeon and Kim for a retrospective study using the CGPaste between 2017 and 2018.  When this study was accepted, was it considered as a Phase I safety study for the 7 patients.  If so this study could be better described as a Phase I/II study with more patients assessed by the need of the >80 to achieve statistical difference for diverse chronic wounds. Show the statistical assessment that was done for determining the inclusion number.

  • We included the information of the statistical assessment in Result section

-The sample size for this clinical trial was calculated as 90 patients (45 patients per group).

According to the literature, the chronic wound area of the control group after 12 weeks was expected to decrease by about 84% compared to the baseline [Ref-2], and the test group was expected to decrease by about 98% [Ref-4].

Therefore, in this study, it was assumed that reasonable clinical improvement was reached when the difference between the groups in wound area reduction rate was 15% or more. (80% power and 5% significance level).

A total of 78 patients (39 patients per group) were calculated according to the above assumptions, but an additional 15% dropout rate was considered.​

Granulation evaluation will need to be described with the full method of evaluation

Epithelization will need to be described with the full method of evaluation

-We included the information

- Granulation tissue formation were evaluated as what percentage of the wound surface was covered with bright, red healthy granulation tissue. Wound epithelization rate were also evaluate as what percentage of the wound surface are epithelized.

Results

Each Case report could be presented in one figure with the photos shown together over time.

Figures 6, 7 and 8 combined

Figures 9, 10, and 11 combined

Figures 12, 13, 14 combined

Or to put all the patient data in one total figure as this would be easily illustrated this way.

-We were guided to upload each figure as separate files. We suppose Journal will edit the figures.

Discussion

Throughout the discussion, the generic use of ADM is not advisable, as each formulation should be described for each study.  Not all formulations are equal and these will need to be described for each study referenced in the discussion.

-In discussion, we tried to separate the information of generic ADM and paste-type ADM, which we used.

However, we did not mentioned the product name of paste-type ADM, as there are other products of ADM manufactured as paste or glue type.

Reviewer 3 Report

Introduction

Please describe better the phases of tissue repair, the difference between typical and atypical ulcers,acute and chronic wounds , the meaning of standard treatment (Wound bed preparation and TIME and management of wound bed and surrounding skin ). 

Material and methods

Please clarify which was your standard treatment in the control group. Did you use debridement? Which dressings did you use and how many times? Did you  bendages and which types?Did you differenciate acute and chronic wounds?(case 1 and 2 are acute wounds and case 3 chronic)

Author Response

Introduction

Please describe better the phases of tissue repair, the difference between typical and atypical ulcers,acute and chronic wounds , the meaning of standard treatment (Wound bed preparation and TIME and management of wound bed and surrounding skin ). 

  • In introduction, we included the information of chronic wounds in detail, as chronic wound is the concern in this study
  • - Wound healing progresses systematically through inflammation, proliferation, and remodeling phases.1,2 Interference in this well-coordinated process, especially in the inflammatory stage, leads to chronic non-healing wounds.2 Chronic wounds often occur in patients with comorbidities such as diabetes, vascular problems (including arterial disease and venous ulcers), or chronic inflammation (such as osteomyelitis, autoimmune disease, and radiation ulcers).3-5

Material and methods

Please clarify which was your standard treatment in the control group. Did you use debridement? Which dressings did you use and how many times? Did you  bendages and which types?Did you differenciate acute and chronic wounds?(case 1 and 2 are acute wounds and case 3 chronic)

-As we mentioned in Material and Methods section,  all patients underwent debridement. Afterwards, the patients were randomized into two groups.

-“Patients were evaluated during a screening period up to 7 days before baseline surgical debridement and treatment. All patients underwent debridement until healthy, viable tissue was visible in the wounds. After surgical preparation of the wound site, patients were randomized into either the experimental group (paste-type ADM and conventional dressing) or control group (conventional dressing) using sequentially numbered, opaque, sealed envelopes, to avoid selection bias. “

- Patients > 19 years old, with chronic wounds with the wound depth ranging from full-thickness skin to bone exposure measuring more than 4 cm2, and failing to heal during a minimum of 3 weeks before the study,9,10,16 were eligible for inclusion.

Therefore, all the patients had chronic wounds.

Round 2

Reviewer 1 Report

The manuscript is better from the initial form even if more than 100 patients should have been included. 

Overall the study is good.

Author Response

Thank you for the comment.

We included statistical evidence of sample size in Result section.

-The sample size for this clinical trial was calculated as 90 patients (45 patients per group).

According to the literature, the chronic wound area of the control group after 12 weeks was expected to decrease by about 84% compared to the baseline [Ref-2], and the test group was expected to decrease by about 98% [Ref-4].

Therefore, in this study, it was assumed that reasonable clinical improvement was reached when the difference between the groups in wound area reduction rate was 15% or more. (80% power and 5% significance level).

A total of 78 patients (39 patients per group) were calculated according to the above assumptions, but an additional 15% dropout rate was considered.​

Reviewer 2 Report

Author's Notes

Line 34. Wound healing progresses instead of wound healing progress

-We changed the word

Line 40. Experiences and not experiences (population may be taken as plural)

-We changed the word

Line 40. Quality-of-life

-We changed the word

Line 48. Not really the relevant references.  Are there any references that are directly related to ADM used in the current clinical study.  More specific information would allow to easily follow the manuscript.

-The reference is related to next paragraph, which explains various application of ADM.

Line 56.  Again, the references do not relate to the phrases and it would be good to assure that other studies and preparations are explained so that there is a clear understanding of the formulation used for each study.

-Reference 12-16 are the studies using ADM.

Materials and Methods

Begin with a summary of the ADM preparation so that it is clear what the formulation of test is composed of, how it was composed (whether all under GMP or in a transplantation program) and under what regulatory (i.e. medical device etc.). Information on ADM (CGPaste; CGBio, 97 Hwaseong, Korea and its registration with a short summary of formulation would be appropriate to include.

Also, the same for EasyFoam, CGBio- whether this is a registered product and could be labelled as such.

-Thank you for the comment. We included the information of the products in Materials and Methods section.

- Two products are used in this study; CG-PASTE (CG Bio Co., Ltd., Seoul, Republic of Korea) is paste-type micronized acellular dermal matrix that is currently used safely in clinical practice. This product has been approved as a medical device applicable to open wounds except for 3-degree burns. Easyfoam (CG Bio Co., Ltd., Seoul, Republic of Korea) is a wound dressing applied to wounds with exudates and protects wounds by maintaining a moist environment on the wound.

Patient

It could be more clear if this is a continuation of the study published in Archives of Plastic Surgery by Jeon and Kim for a retrospective study using the CGPaste between 2017 and 2018.  When this study was accepted, was it considered as a Phase I safety study for the 7 patients.  If so this study could be better described as a Phase I/II study with more patients assessed by the need of the >80 to achieve statistical difference for diverse chronic wounds. Show the statistical assessment that was done for determining the inclusion number.

Again, please put this study into the context of how the first clinical study was done (Phase I) and how this one was designed.  If the products are already on the market and used in the clinical practice, why this study was of interest or necessary….

  • We included the information of the statistical assessment in Result section

-The sample size for this clinical trial was calculated as 90 patients (45 patients per group).

According to the literature, the chronic wound area of the control group after 12 weeks was expected to decrease by about 84% compared to the baseline [Ref-2], and the test group was expected to decrease by about 98% [Ref-4].

Therefore, in this study, it was assumed that reasonable clinical improvement was reached when the difference between the groups in wound area reduction rate was 15% or more. (80% power and 5% significance level).

A total of 78 patients (39 patients per group) were calculated according to the above assumptions, but an additional 15% dropout rate was considered.​

Granulation evaluation will need to be described with the full method of evaluation

Epithelization will need to be described with the full method of evaluation

-We included the information

- Granulation tissue formation was evaluated as what percentage of the wound surface was covered with bright, red healthy granulation tissue. Wound epithelization rate were also evaluate as what percentage of the wound surface are epithelized.

It would be good to include the SOP review-meaning whether there were two or three individual trained clinicians who made independent observations, whether it was done in real-time or with photographs of the patient to do randomly, etc.   This should include the details of the observations to allow to do semi-quantitative analysis.

Results

Each Case report could be presented in one figure with the photos shown together over time.

Figures 6, 7 and 8 combined

Figures 9, 10, and 11 combined

Figures 12, 13, 14 combined

Or to put all the patient data in one total figure as this would be easily illustrated this way.

-We were guided to upload each figure as separate files. We suppose Journal will edit the figures.

This if not what was asked.  A Figure needs to be made by you for each of the suggestions above and not to just upload each photo. 

Discussion

Throughout the discussion, the generic use of ADM is not advisable, as each formulation should be described for each study.  Not all formulations are equal and these will need to be described for each study referenced in the discussion.

-In discussion, we tried to separate the information of generic ADM and paste-type ADM, which we used.

However, we did not mentioned the product name of paste-type ADM, as there are other products of ADM manufactured as paste or glue type.

Changes are not highlighted. It is not clear that the ADM is used as a generic term and how many individual applications in the literature have this term.  There needs to be more descriptive of the past uses of this term and the formulations.

Author Response

Please see responses in Blue.

Author's Notes

Line 34. Wound healing progresses instead of wound healing progress

- As following the reviewer’s recommendation, we changed the word as suggested.

Line 40. Experiences and not experiences (population may be taken as plural)

- As following the reviewer’s recommendation, we changed the word as suggested.

 Line 40. Quality-of-life

- As following the reviewer’s recommendation, we changed the word as suggested.

 Line 48. Not really the relevant references. Are there any references that are directly related to ADM used in the current clinical study. More specific information would allow to easily follow the manuscript.

- Description in line 48 is not about currently used ADM, it is about the conventional type(sheet type) ADM. In general, previous ADM products have been used for skin regeneration as dermal substitutes rather than used for wound regeneration. Therefore, I think those references are appropriate enough to refer general use of ADM. The description of the currently used paste-type ADM is in the next paragraph.

Line 56.  Again, the references do not relate to the phrases and it would be good to assure that other studies and preparations are explained so that there is a clear understanding of the formulation used for each study.

- As following the reviewer’s recommendation, we added related references that used paste-type ADM, but not sheet-type ADM. The formulation of paste-type ADM is not special, it contains ADM and water as the main components, and adjuvants can be varied following the manufacturer’s marketing strategy. In the case of CGpaste, a tiny amount of gelatin was added as an adjuvant. The exact mixing ratio of each component is cannot be published due to the manufacturer’s own patent.

Materials and Methods

 Begin with a summary of the ADM preparation so that it is clear what the formulation of test is composed of, how it was composed (whether all under GMP or in a transplantation program) and under what regulatory (i.e. medical device etc.). Information on ADM (CGPaste; CGBio, 97 Hwaseong, Korea and its registration with a short summary of formulation would be appropriate to include.

Also, the same for EasyFoam, CGBio- whether this is a registered product and could be labelled as such.

-Thank you for the comment. We included the information of the products in Materials and Methods section.

- Two products are used in this study; CG-PASTE (CG Bio Co., Ltd., Seoul, Republic of Korea) is paste-type micronized acellular dermal matrix that is currently used safely in clinical practice. This product has been approved as a medical device applicable to open wounds except for 3-degree burns. Easyfoam (CG Bio Co., Ltd., Seoul, Republic of Korea) is a wound dressing applied to wounds with exudates and protects wounds by maintaining a moist environment on the wound.

Patient

It could be more clear if this is a continuation of the study published in Archives of Plastic Surgery by Jeon and Kim for a retrospective study using the CGPaste between 2017 and 2018.  When this study was accepted, was it considered as a Phase I safety study for the 7 patients.  If so this study could be better described as a Phase I/II study with more patients assessed by the need of the >80 to achieve statistical difference for diverse chronic wounds. Show the statistical assessment that was done for determining the inclusion number.

Again, please put this study into the context of how the first clinical study was done (Phase I) and how this one was designed.  If the products are already on the market and used in the clinical practice, why this study was of interest or necessary….

  • We included the information of the statistical calculation for sample size in Result section as below;

According to the literature, the chronic wound area of the control group after 12 weeks was expected to decrease by about 92.3% compared to the baseline [Ref-2], and the test group was expected to decrease by about 98% [Ref-4]. Therefore, in this study, it was assumed that reasonable clinical improvement was reached when the difference between the groups in wound area reduction rate was 6% or more. (80% power and 5% significance level). The sample size for this clinical trial was calculated as 84 patients (42 patients per group) with considering 15% of drop-out rate. 

The reason why we started the current study is the necessity of randomized controlled trials to prove the efficacy and safety of the product (CGpaste), especially for patients who suffered from chronic wounds. The sheet-type ADM has been widely used as a dermal substitute for more than three decades, but the use of micronized ADM is first reported as a retrospective study on only 12 patients in 2009. In Korea, the first micronized ADM product CGpaste has launched on the market in early 2016, to use for the healing of open wounds. In 2017, when we designed the current study, there were only a few reports that described the efficacy and safety of micronized ADM, especially for the healing of chronic wounds such as DMF and pressure ulcers. Until that time, the NPWT dressing was usually used to achieve granulation of wounds for patients who have chronic wounds. Therefore, the comparative studies had been designed to use NPWT or NPWT with micronized ADM. However, it requires specific equipment for NPWT which makes it uncomfortable when patients move. Therefore, we focused on the establishment of an easier treatment protocol, conventional polyurethane foam dressing was selected as a basic dressing method with or without ADM to treat relatively small size chronic wounds.

Granulation evaluation will need to be described with the full method of evaluation

Epithelization will need to be described with the full method of evaluation

It would be good to include the SOP review-meaning whether there were two or three individual trained clinicians who made independent observations, whether it was done in real-time or with photographs of the patient to do randomly, etc.   This should include the details of the observations to allow to do semi-quantitative analysis.

- We included the information as the reviewer’s recommendation as below.

Wound granulation and epithelialization were evaluated by using photographs of the wounds taken under similar conditions (distance, brightness, etc.). The evaluation of photographs was performed by two independent evaluators who are experts in the plastic surgery department. Granulation tissue formation was evaluated as what percentage of the wound surface was covered with bright, red healthy granulation tissue. Granulation rate was defined as the percentage of the patients who achieved over 75% of granulation by a total number of patients. Epithelialization is defined as the wound covered with an epithelial surface. Epithelialization rate was calculated as the percentage of the patients who achieved epithelization by a total number of patients. Adverse events, including wound infection or any complications, were also evaluated.

Results

Each Case report could be presented in one figure with the photos shown together over time.

Figures 6, 7 and 8 combined

Figures 9, 10, and 11 combined

Figures 12, 13, 14 combined

Or to put all the patient data in one total figure as this would be easily illustrated this way.

This if not what was asked.  A Figure needs to be made by you for each of the suggestions above and not to just upload each photo. 

- As following the reviewer’s recommendation, we combined the photographs from same patient and will upload separated files for three figures (figure 6-8).

Discussion

Throughout the discussion, the generic use of ADM is not advisable, as each formulation should be described for each study.  Not all formulations are equal and these will need to be described for each study referenced in the discussion.

Changes are not highlighted. It is not clear that the ADM is used as a generic term and how many individual applications in the literature have this term.  There needs to be more descriptive of the past uses of this term and the formulations.

- First of all, since more than 2000 published articles were found that used this term (ADM, acellular dermal matrix) in Pubmed until now, the use of "ADM" as a generic term think that acceptable enough. The “acellular dermal matrix” means the product manufactured using cadaver skin with minimal manipulation to remove cells inside. Therefore, the ADM products have very similar components that in the human skin, even though those products are manufactured by different processes. Since the first ADM product was developed as a sheet-type, "ADM" usually means sheet-type products.

The difference between sheet-type and paste-type ADM is only physical properties including viscosity and shape. The development of paste-type ADM was to increase ease of use, especially for irregular-shaped or tunneling wounds. Although the other studies were used different formulations from ours to make paste-type ADM products, the difference does not affect to explain the efficacy and safety of the paste-type ADM for wound healing. Because the ADM is the major component of those products, which acts as an ECM provider during the wound healing process.

Due to the reasons described above, we did not consider the change of “ADM” wording.
